

# Ionospheric density depletions around crustal fields at Mars and their connection to ion frictional heating

Hadi Madanian[1], Troy Hesse[1], Firdevs Duru[2], Marcin Pilinski[1], Rudy Frahm[3]

[1]Laboratory for Atmospheric and Space physics, University of Colorado Boulder, Boulder, CO 80303, USA
[2]Coe College, Cedar Rapids, IA 52402, USA
[3]Space Physics and Engineering Division, Southwest Research Institute, San Antonio, TX 78238, USA

*Correspondence to*: Hadi Madanian (hmadanian@gmail.com)

**Abstract.** Mars' ionosphere is formed through ionization of the neutral atmosphere by solar irradiance, charge exchange, and electron impact. Observations by Mars Atmosphere and Volatile EvolutioN (MAVEN) spacecraft have shown a highly

dynamic ionospheric layer at Mars impacted by loss processes including ion escape, transport, and electron recombination. The crustal fields at Mars can also significantly modulate the ionosphere. We use MAVEN data to perform a statistical analysis of ionospheric density depletions around crustal fields. Events mostly occur when the crustal magnetic field are radial, outward, and with a mild preference towards east in the planetocentric coordinates. We show that events near crustal fields are typically accompanied by an increase in suprathermal electrons within the depletion, either throughout the event or

as a short-lived electron beam. However, no correlation between the changes in the bulk electron densities and suprathermal electron density variations is observed. Our analysis indicates that the temperature of the major ionospheric species, $O_2^+$, increases during most of the density depletion events, which could indicate that some ionospheric density depletions around crustal fields are a result of ion frictional heating.

## 1 Introduction

The solar extreme ultraviolet (EUV) and X-ray radiations at Mars can ionize neutral species ($CO_2$, $O$, $N_2$) to create ionospheric ions and suprathermal photoelectrons (Schunk and Nagy, 2009). Suprathermal photoelectrons and precipitating solar wind electrons can initiate further ionization through impacts with neutrals and cascade in energy to eventually form the bulk electron gas in the ionosphere. Through a series of chemical reactions, $O_2^+$ and $O^+$ become the most dominant ion species at Mars' ionosphere at altitudes above ~200km, with $O^+$ having a longer scale height and requiring a smaller escape

energy (Benna et al., 2015; Fowler et al., 2022; Haider et al., 2011; Withers et al., 2019). The peak of the dayside ionospheric layer at Mars is typically formed at around 110-150 km altitudes (Girazian et al., 2020; Vogt et al., 2017). Lack of a dipole field at Mars and the short stand-off distance of the bow shock from the surface makes the ionosphere highly susceptible to upstream effects. The ionosphere is highly variable, changing with the amount of solar flux which varies by the solar zenith angle and seasons, and variations of upstream solar wind conditions.



The Martian ionosphere can also be modulated by Mars residual crustal magnetic fields that are mostly present in the southern hemisphere (Dubinin et al., 2016; González-Galindo et al., 2021; Withers et al., 2016). Near the crustal fields and due to reduced tailward transport rate, $O^+$ ions seem to linger on the dayside where they are generated (Lundin et al., 2011). Strong crustal fields can also trap low energy ions and reduce the ion pickup and escape rate around these fields (Fan et al., 2019). Effects of crustal fields on electrons is more variable and less determined. Statistical studies have shown that
electrons trapped on closed crustal field lines in general have longer lifetime and exhibit higher densities and lower temperatures compared to other places and these effects increase with altitude and are modestly affected by the solar wind conditions (Andrews et al., 2023; Flynn et al., 2017). In general, elevated plasma densities are observed near crustal fields (Andrews et al., 2015). On the nightside, suprathermal electrons that can ionize the ionosphere have limited access to the atmosphere around closed crustal magnetic fields resulting in smaller ion densities near crustal fields (Girazian et al., 2017).
The ionopause, which appears as a steep gradient in the bulk electron density altitude profile data, is also suppressed around strong crustal fields as these fields limit the access of precipitating electrons (Chu et al., 2019).

Sudden ionospheric density depletions at Mars have been the topic of several previous studies. In essence, density depletions are sudden decreases in the plasma density altitude profiles that are inconsistent with the average. Several statistical studies have indicated that although these depletions are observed across the entire Martian ionosphere, there is a tendency for
observing these dropouts near the crustal fields (Basuvaraj et al., 2022; Duru et al., 2011; Withers, 2005). Early investigations on the topic using Mars Global Surveyor Radio Science experiment discussed depletions or "bite-outs" observed in anomalous ionospheric density profiles that were mainly observed near crustal magnetic fields at Mars (Withers, 2005). It was suggested that more analysis of these events is necessary to determine the nature of these events. Analysis of data from Neutral Gas and Ion Mass Spectrometer (NGIMS) (Mahaffy et al., 2015b) on the MAVEN spacecraft around these
structures has shown that these electron deficit structures are bubble-like in shape and seem to be more frequent on the nightside (Basuvaraj et al., 2022). Duru et al. [2011] used data from Mars Advanced Radar for Subsurface and Ionospheric Sounding (MARSIS) instrument onboard the Mars Express spacecraft to analyze these structures on the nightside and near the terminator regions to show that some events are aligned with the onset of a photoelectron boundary. In a study of these depletions on the nigh side of Mars using the same dataset, (Cao et al., 2022) argued that large amplitude depressions of the
total electron content near strong crustal fields could be, although not always, related to shielding of precipitating suprathermal electrons, while small amplitude depletions of bulk electrons could occur anywhere across the ionosphere. The cooccurrence (or lack thereof) of bulk electron depletion with suprathermal electrons has not been established so far.

A similar nomenclature was used for density structures at Venus' nightside ionosphere which are referred to as ionospheric density holes. The generation mechanism of those structures is associated with the radial extension of draped magnetic field
lines around the planet where the plasma can flow and be depleted tailward (Brace et al., 1982). Certain density ridges and troughs in the Martian ionosphere could be compared to sporadic E-like layers at Earth' ionosphere near the equator where two counter-streaming plasma flows interact causing relative ion drifts (Collinson et al., 2020). However, the region of crustal magnetic fields at Mars in some respect most likely resembles the polar cap regions of Earth where the geomagnetic



fields have radial geometry at low altitudes and become more horizontal farther out. Density depletions in the high-latitude
ionospheric F-layer are commonly observed in incoherent scatter radar data (Bjoland et al., 2021). Such ionospheric troughs
could be caused by enhanced electron dissociative recombination rates driven by warmer ions. Enhancement of the ion
temperature increases the charge exchange rate with neutrals, e.g., $O_2$ and $N_2$ creating ion products that recombine quickly
with electrons leading to consumption of cold electrons and ions (Rodger et al., 1992).

In this paper, we focus our efforts on characterizing density depletions around crustal magnetic fields at Mars and perform a
statistical analysis of plasma properties during these events. Irregular and unexpected depletions of the ionosphere can have
major consequences for ion escape and space weather at Mars (González-Galindo et al., 2021; Xu et al., 2020). As such, it is
important to investigate and characterize these events. The manuscript is organized as follows: in section 2 we introduce the
data sources and our analysis methods and approach for event selection. Section 3 contains our observation results. We
discuss the results in Section 4 and conclusions are provided in Section 5.

## 2 Methodology, Event Selection, and Data Sources

In this study, we use data from the MAVEN spacecraft. We surveyed Langmuir Probe and Wave (LPW) (Andersson et al.,
2015) measurements between 2015 up to 2022 for ionospheric density depletions. Due to highly dynamic and turbulent
Martian ionosphere, automated detection algorithms result in large number of false flags. As such, we identified these events
visually and when the density measurements exhibit sharp depletions both in timeseries and altitude profile data. To quantify
the closeness of a density profile measurement to the crustal fields and the level of influence from the crustal fields on each
event we define the proximity parameter $\zeta$:

$$\zeta = \sum_{i=x,y,z} \frac{\Sigma |B_{i,sc} - B_{i,m}|}{\Sigma |B_{i,m}|} \qquad (1)$$

where $B_{i,sc}$ is the spacecraft measurements of the magnetic field and $B_{i,m}$ is the crustal magnetic field modeled at the
position of the spacecraft (Morschhauser et al., 2014). $\zeta$ provides an estimate of the crustal field prevalence or a measure of
closeness of the observations to the crustal fields at Mars. Lower $\zeta$ values indicate more contributions from the crustal field
with $\zeta = 0$ meaning observed fields are identical to model predictions. Contributions of external magnetic fields from the
solar wind or perturbation to the field generated locally by instabilities or currents lead to increased deviation of observations
from the model and higher values of $\zeta$. In deriving Equation 1, effects of generic similarities between time series (constant
arrays), singularities, and absolute strength of the fields are also considered.

Figure 1 shows an example of an ionospheric depletion event on 2 November 2016. Panel (a) shows three components of the
magnetic field as measured by the Magnetometer (MAG, (Connerney et al., 2015)) (markers) and the modeled crustal fields
(solid lines). The area between the curves along each component (numerator in Eq. 1) is used in determining $\zeta$. Panel b



shows the electron density measurements by LPW with a depletion apparent between 04:08:00 and 04:09:28 UT. The
minimum accepted quality flag in LPW data is 50. The density of suprathermal electrons measured by Solar Wind Electron
Analyzer (SWEA, (Mitchell et al., 2016)) increases during this time as seen in Panel c. Panel d indicates that densities of
ionospheric heavy ions $O^+$ and $O_2^+$, as measured by Suprathermal and Thermal Ion Composition (STATIC, (McFadden et
al., 2015)) also decrease during this event.

We down select altitude profiles that show sudden density depletions in the profiles whose $\zeta$ is less than 5. This threshold is
set by visual inspection of several events and the total number of events that would go into our statistical analysis to provide
a reasonable sample size. We find 242 events in LPW data set. STATIC ion density data are available for 135 of these
events. $O_2^+$ ion temperatures can be derived from STATIC measurements and are available during 83 events. Limitations
dictated by spacecraft attitude, instrument pointing and ram direction, and ion abundance at a given altitude restrict the
available times when STATIC measurements can be properly calibrated for derivation of $O_2^+$ ion temperatures. For details of
the STATIC calibration process and derivation of ion densities and temperatures readers are referred to the STATIC
instrument paper and follow up calibration studies (McFadden et al., 2015; Hanley et al., 2021; Fowler et al., 2022). SWEA
measurements of suprathermal electrons over the full energy range (3 eV to 4.6 keV) are available during 95 events. This is
mainly due to a change made in SWEA's energy sweep table during 2020 which raised the minimum scan energy.
Nevertheless, available data provide a reasonable sample size to analyze the behavior of suprathermal electrons.

## 3 Observations

The depth of a depletion (or hole) is defined as the ratio of the lowest bulk electron density inside the hole to the highest
electron density on either sides/edges of the depletion. The event distributions along three components of the magnetic field
taken at the core (the lowest density point) of the depletion are shown in Figure 2. The spherical coordinates of the magnetic
field in the Mars body-fixed planetocentric frame are used. In this coordinate system, r is along planet's radius and $\varphi$ and $\theta$
are polar coordinates (parallel to the surface) changing between [0°, 360°] and [-90°, 90°], respectively. By comparing
figures 2a to 2b and 2c we find that the radial component of the field shows the highest level of variations compared to field
components parallel to the surface, with perhaps a slight preference towards +Br, or when the crustal fields exit the surface.
The histogram probability distribution for each field component is also overplotted on each panel.

Panel d shows the depletion depths as a function of $\zeta$ parameter. The depletion depth does not depend on $\zeta$. Data points on
this panel are color coded by the total strength of the magnetic field. Events with the highest magnetic field strengths (purple
dots) also appear at low $\zeta$ (i.e., completely dominated by strong crustal fields) while the depletion depth for these events is
mostly above 0.1.



In Figures 3 we discuss the distribution of $\zeta$ versus normalized field components. The scatter plot of events along $\widehat{B}_\theta$ (a) and $\widehat{B}_\varphi$ (b) with associated probability distribution histograms below each plot. The histogram to the right shows the distribution of $\zeta$ with a peak at around 1. The $\widehat{B}_\theta$ distribution exhibits a peak at 0.2. It is difficult from this plot to determine whether there is any preference on the direction of $\widehat{B}_\theta$. While the event distribution in $\widehat{B}_\varphi$ appears to be more frequent along positive $\varphi$ suggesting that the depletions are found more likely around crustal magnetic fields pointing eastward.

As we discussed in Figure 2a, there is a preference for observing density depletions when the crustal fields have a radial orientation (i.e., exiting the Martian surface). This preference is clearly seen in Figure 4, where the probability distribution of events along $\widehat{B}_r$ is shown. The distribution of events in altitude exhibits two major peaks, one observed at ~ 265 km and the other more concentrated at 410 km. We find no correlation between $\zeta$ values and the altitude at which core of the depletions are observed.


Previous studies have shown that the flux of suprathermal electrons can increase within some depletion events and decrease or even disappear for other events (Cao et al., 2022; Duru et al., 2011; Nielsen et al., 2007). The cause and the underlying process that controls the suprathermal electrons within the ionospheric density depletion is not well understood. Here, by using MAVEN observations of different electron populations (i.e., bulk/thermal and suprathermal) we choose a more

quantitative approach to study the dynamics of suprathermal electrons within depletion events near crustal fields. It appears that events can be divided into two main categories. Some depletion events can be accompanied by enhanced fluxes of suprathermal electrons over almost the entire event period while other events exhibit a short-lived beam-like surge of electrons either in the middle of the depletion or near the boundaries. We define three measures to quantify electron density variations. $\Delta n_{e,C}$ is the change in cold or bulk electron density as measured by LPW, between the lowest density within the

depletion and the measured density outside the depletion. $\Delta n_{e,S} = n_{eS,Max} - n_{eS,edge}$ is the difference between the maximum suprathermal electron density inside the depletion ($n_{eS,Max}$) and the density outside the depletion ($n_{eS,edge}$) and gives an estimate of the increase or decrease in suprathermal electrons, either throughout the depletion or as a sudden pulse. We also define $\delta n_{e,S} = n_{eS,Max} - \tilde{n}_{eS}$ which is the difference between $n_{eS,Max}$ and $\tilde{n}_{eS}$ or the average suprathermal electron density over the depletion period. A higher value of $\delta n_{e,S}$ is indicative of a stronger and more intense suprathermal electron beam.


Figure 5a shows the dependence of $\Delta n_{e,S}$ on $\Delta n_{e,C}$. All depletion events but six are accompanied by an enhancement in the suprathermal electron fluxes. This is perhaps a distinguishing aspect between events near the crustal fields considered in this paper and those discussed in previous studies of depletions across the entire ionosphere for which no clear pattern in the abundance of suprathermal electrons are found. We note that density variations of the suprathermal and bulk electron

populations during depletion events do not seem to be correlated. This provides evidence against an acceleration mechanism that shifts part of the bulk electron population to higher energies while creating a density bite-out. In Figure 5b, variations of $\delta n_{eS}$ versus $\Delta n_{e,C}$ are shown. Data points are colored based on the altitude at which the depletions are observed. We see



many events at low altitudes that exhibit high values of $\Delta n_{e,C}$ ($> 10^3$ cm$^{-3}$) compared to events at other altitudes, suggesting that stronger beams of suprathermal electrons more likely occur at lower altitudes, where crustal fields are stronger.

As data in Figure 1d suggests, ionospheric ions follow a similar depletion pattern to that of bulk electrons during these events. In Figure 6, we analyze the density variations of the most abundant ionospheric ion species, $O^+$ and $O_2^+$. There is a shift towards higher $O_2^+$ variations with increase in $\Delta n_{e,S}$ (the Colorbar). That is because at lower altitudes where bulk electron density variations are higher, variations in $O_2^+$ densities also become more dominant because $O_2^+$ is the dominant ion species at low altitudes and has a shorter scale height compared to $O^+$. Our interpretations of data shown in this figure are
consistent with previous studies that analyzed ion densities using NGIMS measurements (Basuvaraj et al., 2022).

Figure 7 shows another example of a density hole in Martian ionosphere. LPW bulk electron and STATIC ion density measurements are shown in panels a and d respectively. We also show the temperatures of bulk electrons, suprathermal electrons, and ions in panels b, c, and e as measured by LPW, SWEA and STATIC instruments. These panels show that the
temperatures of bulk electrons and ions increase during the depletion while the suprathermal electron temperature decreases. The electron plasma parameters in a Langmuir probe are obtained from the probe characteristic current-voltage curves. The slope of this curve, which is inversely proportional to the electron temperature, could decrease due to reduced rate of change in the electron saturation current. However, that current can decrease if a plasma sink is actively present in the plasma (as is the case for density holes,) rather than an actual change in the temperature of the electron gas. The temperatures derived
from measurements of electrostatic analyzers (SWEA, STATIC) on the other hand are determined from the energy extent (width) of the particles entering the instrument and thus directly relate to the average temperature of charged particles. It is therefore likely that the increase in the bulk electron temperature is an artifact of sudden depletion of the cold electron gas within the density hole which leaves only "warm" electrons to be probed by the LPW, rather than heating of electrons by a physical process and increase in their thermal velocity.


$O_2^+$ ion temperatures in Figure 7e show that ion temperatures within the density depletion have higher temperatures than the surrounding plasma. The higher ion temperature could be a sign of ion frictional heating which arises due to a relative ion drift in the frame of neutral species. External electric fields such as the convection electric field, or atmospheric waves and other dynamic processes can cause a velocity difference between ions and neutral. The electric field creates an $\mathbf{E} \times \mathbf{B}$ drift in
the plasma which increases the relative velocity between ions and neutrals, while the motion of the background atmospheric neutral species relative to ions also leads to similar velocity drifts. In either case, at large enough relative drift velocities frictional heating occurs which increases the ion temperatures.

As shown in Figure 8, out of 83 events with full or partial ion temperature estimates, three events exhibit a reduction in ion temperatures inside the depletion. Since we investigate density depletions, the number of events with available temperature
data decreases. The decreased ion abundance further complicates the temperature derivations as a minimum ion count rate is required for reliable determination of the temperatures. Figure 8 also indicates that events clustered at around 400 km



altitude show more drastic variations in ion temperatures, which could be associated with increased heating of ions due to their drift relative to neutrals. Such a velocity difference typically increases with altitude, while the heating process itself is bound by the abundance of the local neutral species.


## 4 Discussion

Physical processes that cause the ionospheric density depletions at Mars are not well understood. By studying these events near crustal magnetic fields, we focus our attention to generation mechanisms associated with effects that can be mapped along crustal magnetic field lines and effects that can impact charged particles at higher altitudes. Ion pickup is a major ion loss process at high altitudes around Mars (Cravens et al., 2002). We show in Figures 1 and 6 that ions with different masses are depleted during these events indicating that ion pickup cannot explain the loss of these ions as this process is much slower for heavier ions ($O_2^+$ in this case). Electrodynamical forces in the ionosphere tend to have opposite impacts on ions versus electrons.

In the Martian upper atmosphere, $CO_2$ remains the most abundant neutral species up to around ~200 km altitude, above which atomic oxygens becomes the dominant neutral constituent with lower abundances of $N_2$ and $O_2$ (Benna et al., 2015; Mahaffy et al., 2015a). At altitudes above 200 km, $O_2^+$ and $O^+$ become the most abundant ionospheric ion species. Chemical loss of $O^+$ ions is a slow process. This is mainly due to due to the small electron recombination and ion-neutral reaction rate coefficients. Some of the relevant reactions area listed below (Schunk and Nagy, 2009):

$$O^+ + N_2 \rightarrow NO^+ + N \qquad 1.2 \times 10^{-12} \text{ cm}^3\text{s}^{-1} \qquad (2)$$

$$O^+ + O_2 \rightarrow O_2^+ + O \qquad 2.0 \times 10^{-11} \text{ cm}^3\text{s}^{-1} \qquad (3)$$

$$O^+ + CO_2 \rightarrow O_2^+ + CO \qquad 1.1 \times 10^{-9} \text{ cm}^3\text{s}^{-1} \qquad (4)$$

$$O_2^+ + NO \rightarrow NO^+ + O_2 \qquad 4.6 \times 10^{-10} \text{ cm}^3\text{s}^{-1} \qquad (5)$$

$$O^+ + e \rightarrow O \qquad 1.76 \times 10^{-10} T_e^{-0.7} \qquad (6)$$

$$O_2^+ + e \rightarrow O + O \qquad 1.3 \times 10^{-5} T_e^{-0.7} \qquad (7)$$

$$NO^+ + e \rightarrow N + O \qquad 6.93 \times 10^{-6} T_e^{-0.5} \qquad (8)$$

While the reaction of $O^+$ ions with neutral species is slow, the byproducts of reactions in Equations 2 – 4, namely $O_2^+$ and $NO^+$ ions, can quickly recombine with bulk electrons, removing both ions and electrons from the plasma in the process. Recombination of $O_2^+$ ions with electrons is orders of magnitude faster than $O^+$. In addition, the ion-neutral reaction rate coefficients are energy dependent and increase with ion temperature (Rodger et al., 1992; St.-Maurice and Torr, 1978; Viggiano et al., 1992). For instance, Figure 8 in Rodger et al. (1992) shows that the reaction in Equation 3 can be an order of magnitude faster for an ion temperature increase of ~8,000 K (or 0.68 eV). Frictional heating of ions increases the ion



temperatures which in turn results in increased ion-neutral reaction rates and production of ions that recombine faster with electrons and create localized density depletion zones near crustal fields.

Data in Figures 7 and 8 indicate that the temperature of ions clearly increases within the depletion events. The discussion of the exact process (external electric fields or atmospheric disturbances,) that initiate the frictional heating in the ionosphere is left for a future study. But here we note that the ion ($O^+$) gyrofrequency is smaller than the ion-neutral collision frequency, meaning that ions are likely to undergo at least one collision before they can complete a full gyration around the magnetic field and can be considered nonmagnetized, while electrons are magnetized. Electron-ion dissociative recombination

removes caused the density depressions in timeseries data. In return we expect to see an increase in the neutral density. However, since the change in the plasma density is on the order of a few thousands $cm^{-3}$ or less, it is difficult to detect such small variations in a background neutral atmosphere of $10^5 - 10^8$ $cm^{-3}$.

For radial crustal fields, the $\mathbf{E} \times \mathbf{B}$ can only result in horizontal drift and heating of ions parallel to the Martian surface. This could have an effect on these depletions to appear as elongated horizontal structures in spacecraft observations (Basuvaraj et

al., 2022). When exposed to the solar extreme ultraviolet flux, the new neutral products from the dissociative recombination will be photoionized creating new pairs of electrons and ions, which could explain the surge of suprathermal electrons within these structures (Duru et al., 2011).

**5 Conclusion**

In this statistical study, we focus on depletion events observed in the vicinity of crustal fields at Mars. We define the

proximity parameter $\zeta$ to quantify the proximity of observed events to Martian crustal magnetic fields. We use MAVEN observations between 2015 to 2022 to analyze ionospheric density depletions around crustal fields. Ion measurements from STATIC, cold electron measurements from LPW and suprathermal electron measurements from the SWEA instrument around these events are utilized. In our analysis, we use the differences of the ionospheric densities inside and outside the depletion events to quantify these structures. This way, effects of solar zenith angle, seasons, and heliocentric distance on

observed variations are minimized (Andrews et al., 2023).

We investigate variations in different plasma populations within the density depletions. We note that due to the way the spacecraft cuts through a 3-dimensional density structures, the apparent variations (e.g., hole depth, etc.) are independent of the actual depletion depth of the depletion structure. Nevertheless, we find that suprathermal electrons are almost always present within the density depletions near crustal fields. We show that increase in the ion temperature during some events

could be associated with ion frictional heating, which could also be causing the depletion through a two-step process. Heating of ions increases the charge exchange reaction rates, followed by electron dissociative recombination of the ions which remove both electrons and ions creating isolated plasma depletions.



**Data availability**

All data presented in the figures are publicly available on the Planetary Data System https://pds.nasa.gov and mirrored at
http://sprg.ssl.berkeley.edu/data/maven/data/sci/. LPW density and temperatures are available through
pds://PPI/maven.lpw.derived/data/lp-nt. STATIC density and temperature data can be accessed at
http://sprg.ssl.berkeley.edu/data/maven/data/sci/sta/l3.

**Author contribution**

HM conceptualized the idea, defined the analyses, and carried them out. TH, FD, and RF performed parts of the data
curation and event selections. MP assisted with NGIMS data analysis. HM prepared the manuscript with contributions from
all co-authors.

**Competing interests**

The authors declare that they have no conflict of interest.

**Acknowledgments**

We acknowledge the support from the MAVEN contract to the Laboratory for Atmospheric and Space Physics (LASP) at the
University of Colorado in Boulder.

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



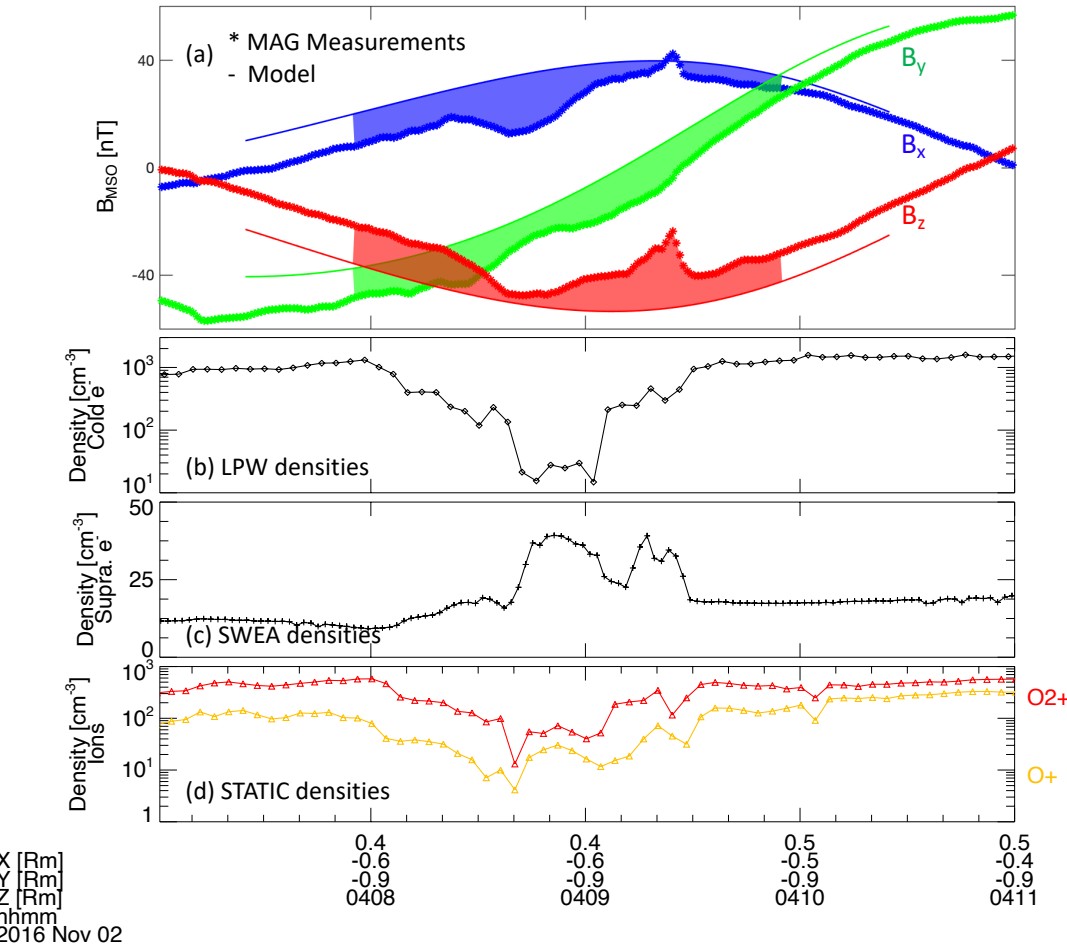

**Figure 1.** (a) Magnetic field measured by the MAVEN magnetometer (asterisks) and modeled crustal fields (solid lines) in the Mars Solar Orbital (MSO) coordinates, (b) bulk electron density measurement by LPW, (c) suprathermal electron density measurements by SWEA, (d) STATIC density measurements of $O^+$ (yellow) and $O_2^+$ (red) ions.



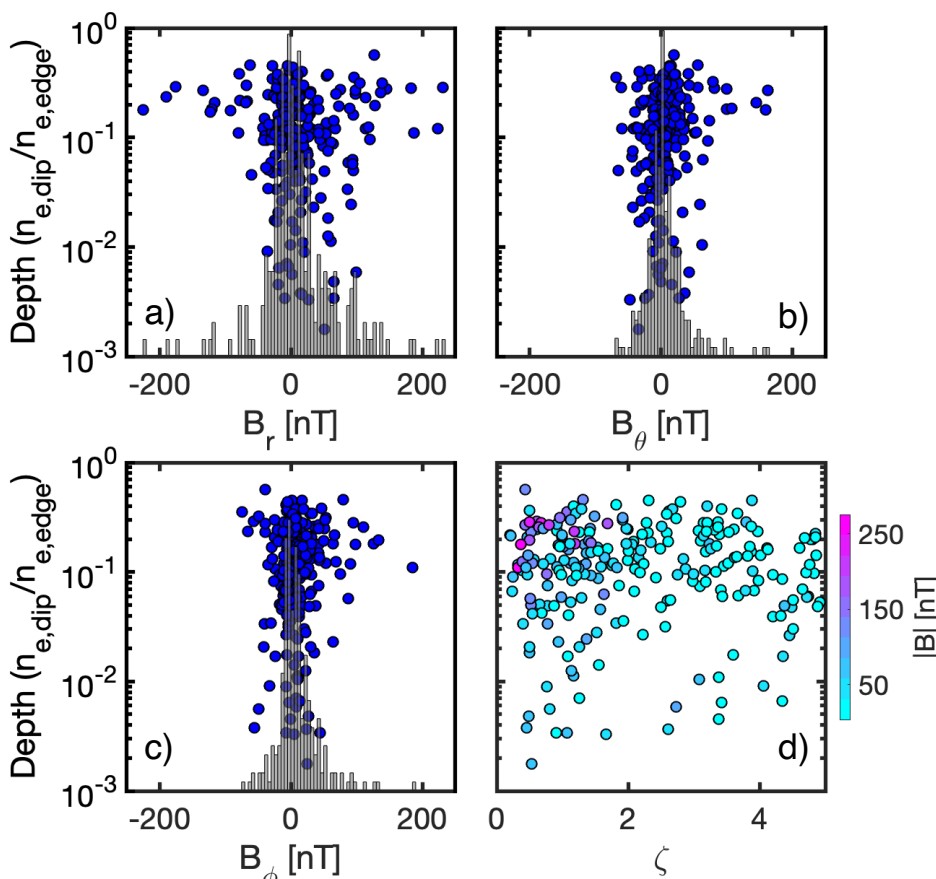

**Figure 2.** Distribution of depletion depth versus a) $B_r$ b) $B_\theta$, c) $B_\varphi$, and d) $\zeta$. Normalized bar plots in grey in the background on panels a - c are the relative occurrence rate of events. Data points on panel d are color coded by the strength of the magnetic field.





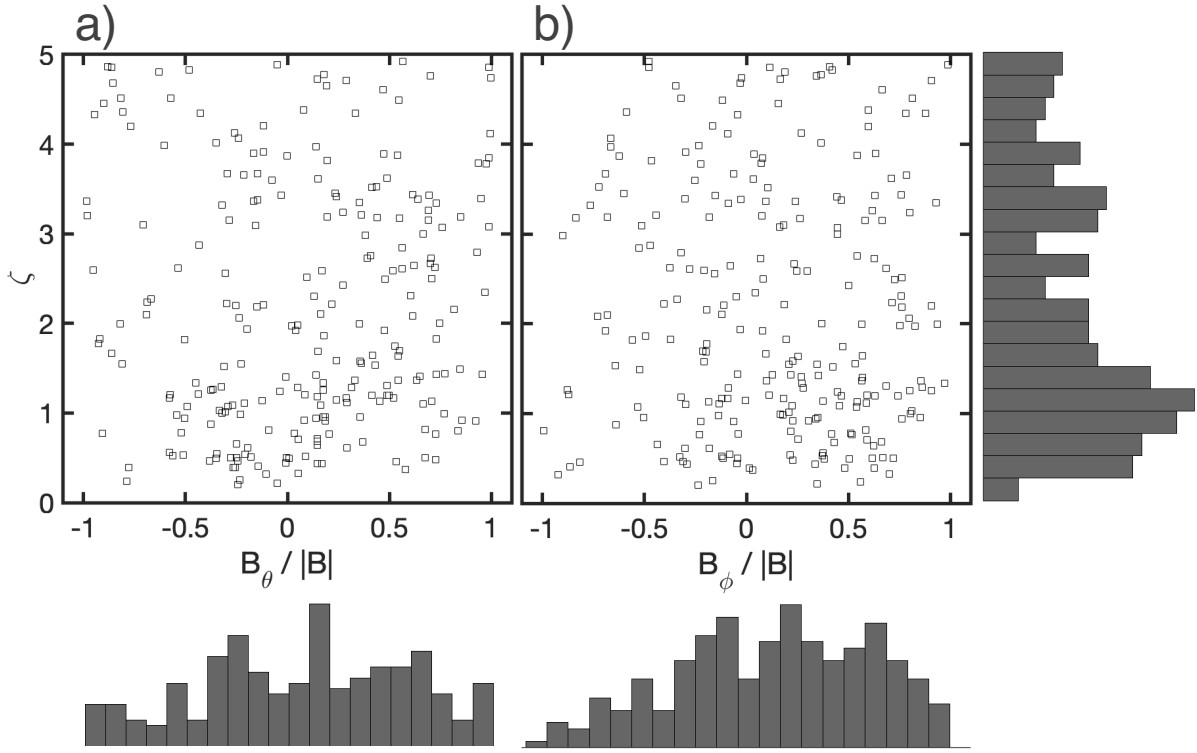

**Figure 3.** Scatter plots of events versus magnetic field unit vector along $\theta$ (a) and $\varphi$ (b). The normalized probability distribution for each component is shown below the corresponding panel, with the distribution for $\zeta$ shown to the right of panel b.




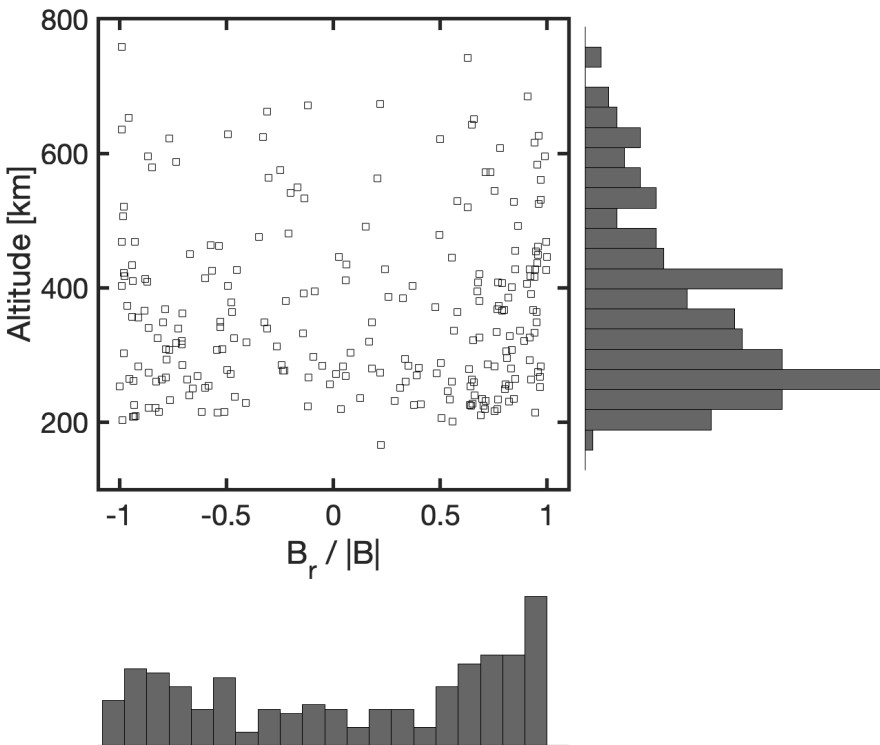

**Figure 4.** Scatter plot of density depletions versus the observed altitude around Mars and the unit vector along the Martian radius. Histograms show the occurrence probability of events.

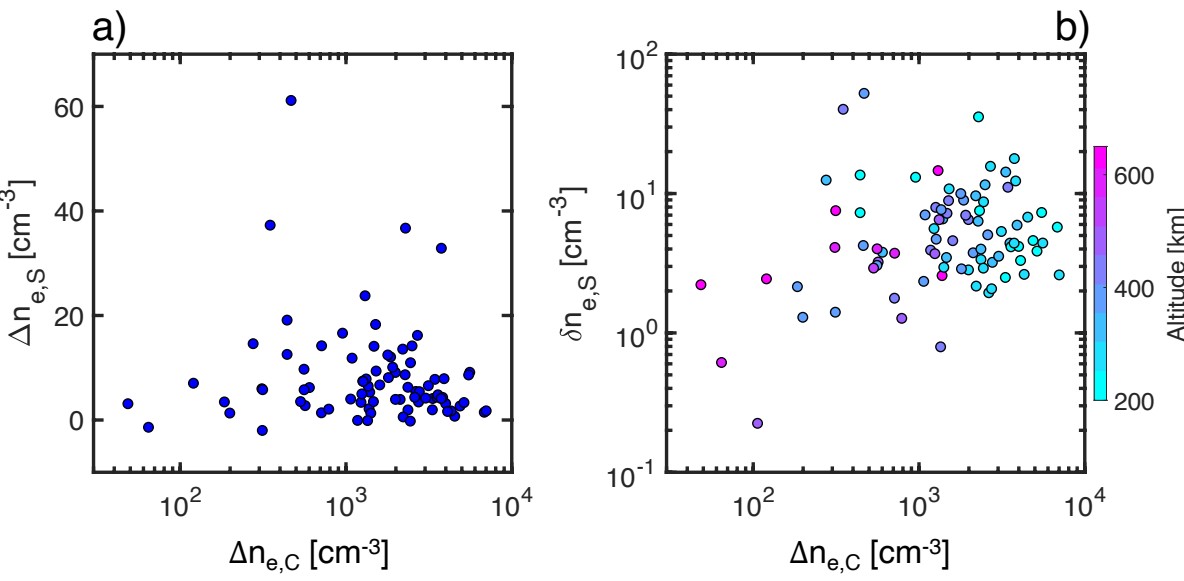



**Figure 5.** Scatter plots of suprathermal electron density variations versus changes in the bulk electron density across the depletions. Panel (a) shows the maximum difference in suprathermal electron density between inside and outside the depletion. Panel (b) shows the difference between the maximum suprathermal electron density and the average suprathermal density inside the depletions. Points in panel b are color coded by event altitudes.


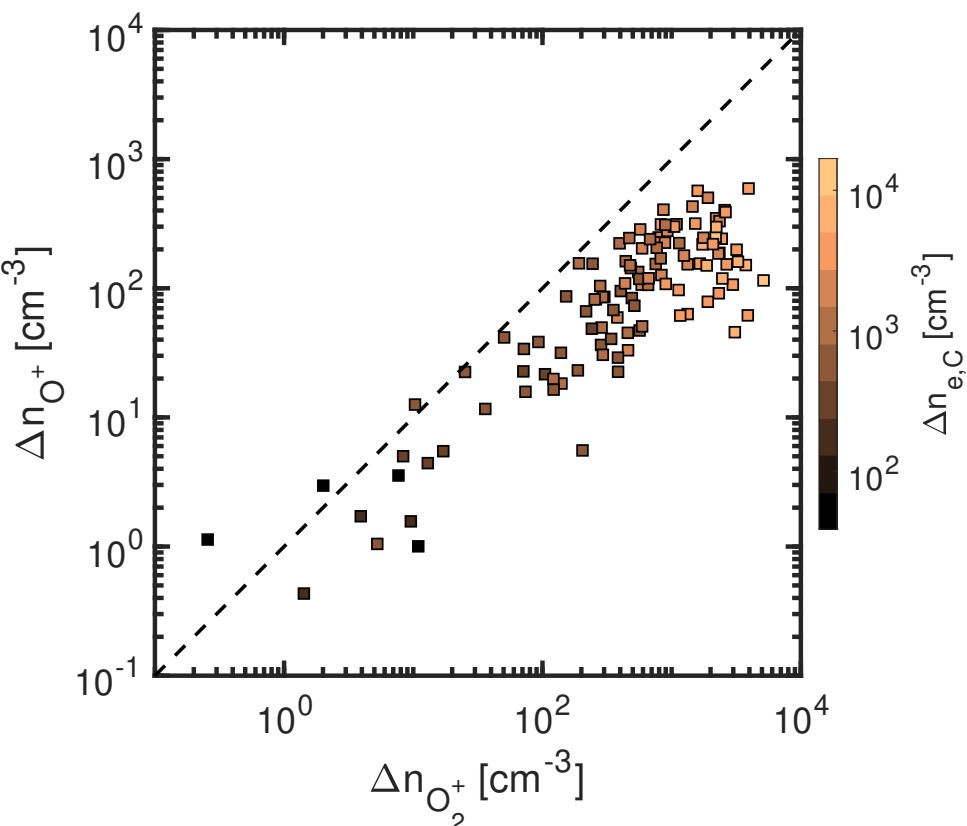

**Figure 6.** Density variations of most abundant cold ionospheric species ($O^+$ and $O_2^+$). Points are color coded by corresponding changes in the bulk electron density. Excluded in the figure are eight depletion events which exhibit increased ion densities within the depletion compared to the ambient plasma.





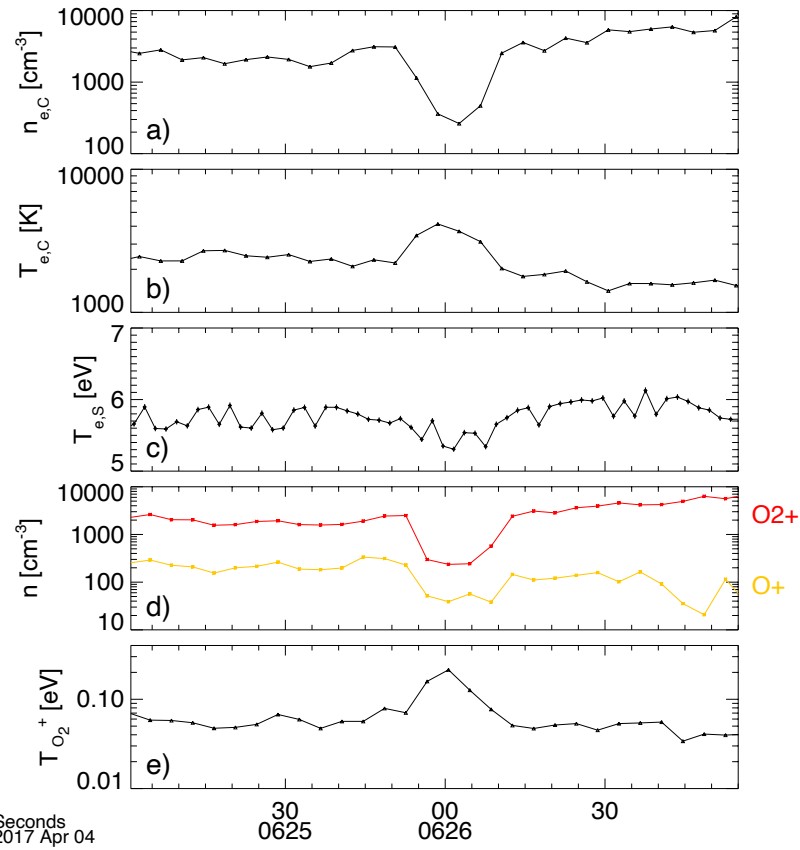

**Figure 7.** Timeseries of a density depletion event on 4 April 2017 at 06:26:00 UT. Panels show (a) the density and (b) the temperature of bulk electrons measured by LPW, (c) the suprathermal electron temperatures measured by SWEA, (d) $O^+$ (yellow) and $O_2^+$ (red) densities, and (e) $O_2^+$ temperatures measured by STATIC.






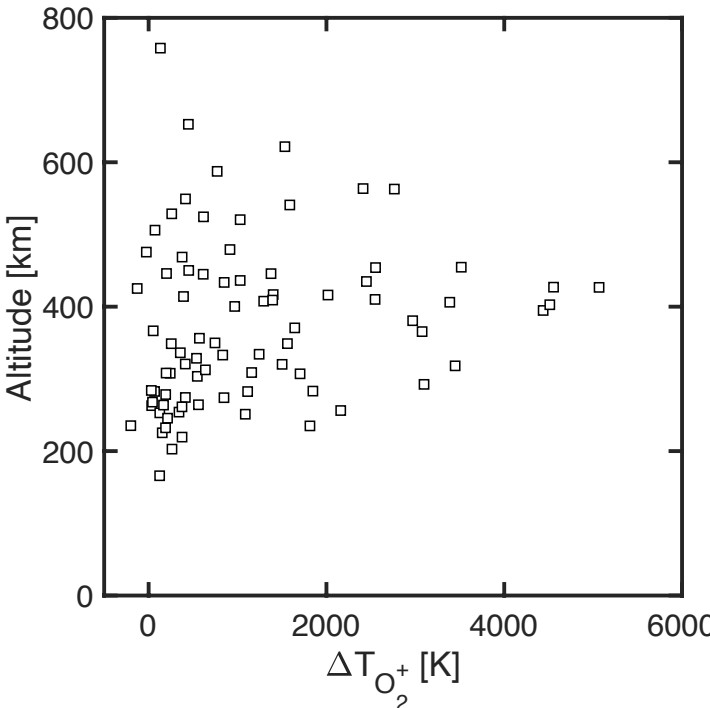

**Figure 8.** Temperature variations, in Kelvin, of $O_2^+$ ions inside the depletions as a function of altitude taken at the middle of the depletion events.