# Peer review of "Ionospheric density depletions around crustal fields at Mars and their connection to ion frictional heating"

_EGUsphere, 2023_

## Referee Comment (RC2)

**Ionospheric density depletions around crustal fields at Mars and their connection to ion frictional heating**

**GENERAL COMMENTS**

The authors investigate how density depletions in the ionosphere of Mars are correlated with the crustal fields and show how said depletions might be connected to ion frictional heating. The paper is well written and presents different aspects of the topic taking advantage of the available MAVEN observations from several instruments. The authors define certain parameters which then use to quantify the correlation between the crustal fields and the ionospheric depletions and their connection to ion frictional heating. The plots are clear and help the reader to understand the main points of the paper. There are some parts in the paper though that need further clarification.

**SPECIFIC COMMENTS**

**2. Methodology, Event Selection, and Data Sources:**

- Could the authors perhaps add a short paragraph about the instruments used in the paper? For example, what each instrument measures and such.
- Do the authors survey the whole LPW data from 2015 to 2022 or a part of it? For example, only dayside, what range of SZAs and altitudes? or have the authors used some other specific criteria? How many orbits do the authors check? Maybe the authors could be more specific here.
- Line 79: Could the authors demonstrate an identification example plot from the data like they show in Figure 1. Perhaps the authors could just add to Fig. 1 a density profile and some lines on the plots indicating where the event starts and ends to show how they identify the depletions. That would also help the readers later when the authors describe the various Δn parameters and refer to measurements near the boundaries or out of the boundaries.
- How do the authors search for the depletions? Do they first check the time series and if there is something they also check the profile? They check both time series and profiles for each orbit and compare? I would like to see a more detailed description of the identification process which would also fit nicely with the recommendation above about demonstrating the identification in a plot.
- When the authors calculate the proximity parameter ζ, and use the magnetic filed measurements they should also state the coordinate system they use, for example that the $B_{i, sc}$ x, y, z components are in the MSO system.
- Lines 89-90: ''In deriving Equation 1, effects of generic similarities between time series (constant arrays), singularities, and absolute strength of the fields are also considered'' → Could the authors elaborate on that? Perhaps they could explain these effects in more detail and how they are considered.

- Line 95: ''The minimum accepted quality flag in LPW data is 50'' → Could the authors elaborate on that? What does that mean exactly? It would be useful for readers who are not familiar with LPW data too.
- Line 100: ''We down select…''→ Could the authors elaborate on that? Events with $\zeta <$ 5 are selected but out of how many and why? Why do they authors choose $\zeta < 5$? Arbitrarily? Can the authors provide a plot like a histogram/distribution with all the $\zeta$ measurements and the crustal fields values to show why they choose 5? Or a plot that shows $\zeta$ as a function of the strength of the crustal fields?
- Lines 101-102: ''…visual inspection of several events…''→ how many events are identified in total? How many were inspected?
- Lines 101-102:''…the total number of events… a reasonable sample size''→ In my opinion this is not the right reason to select the right value for $\zeta$. If there were fewer events for example, would the authors select events with much higher $\zeta$ – and thus farther from crustal fields – just to have a sufficient number to do statistics?
- It would be helpful for the readers if the authors could somehow present either by giving some numbers or with a plot as previously suggested, what happens at $\zeta = 5$. How strong the crustal fields are in the identified depletions for $\zeta = 5$, and what happens below and above this limit.
- Line 102:''We find 242 events in LPW data set.'' → Total events? Events with zeta<5? Also, in what locations the authors see the depletions? Perhaps the authors can include a crustal field map with the locations of the depletions and/or altitude and SZAs of the events.
- Figure 1→ The authors could add the altitude in km and the SZA as well below the figure.

**3. Observations:**

- Lines 112-113: An example here would be helpful. If the authors could add vertical lines for example in Figure 1 with the minimum density and the limits of the hole (left and right) it would be easier to demonstrate exactly how they calculate the depth.
- The authors now use spherical coordinates for the magnetic field. For the $\zeta$ calculation MSO was used. Maybe the authors could emphasize that and also elaborate a little on why spherical coordinates are more appropriate for their analysis.
- Line 120: ''The depletion depth does not depend on $\zeta$ ''→ If I understand correctly, this means that the depletion depth does not depend on the crustal fields. Would be interesting to compare depletions farther from the crustal fields to see if you get the same depth distribution.
- Line 121: ''…color coded by the total strength of the magnetic field.'' → Do the authors mean the total strength measured or modelled (crustal fields) ?
- Line 121-123: ''Events with the highest magnetic…appear at low $\zeta$…mostly above 0.1''→ I am a little confused with that statement. Events with high magnetic field strength would also be the events where crustal fields dominate so by definition $\zeta$ should be low. Also the fact that the depletion depth is larger for those events does it mean that there is after all a correlation between crustal fields and depletion depth? Because the previous statement says that the depth does not depend on $\zeta$.

- Figure 3 – Perhaps the authors could give a more detailed description of the plot. Since the same format is used in Figure 4 as well.
- Line 133: No correlation between $\zeta$ and altitude. So no correlation of the crustal fields and the altitude? Perhaps the authors could emphasize that in the paper.
- Lines 141-143: ''some depletion events…or near the boundaries'' → Perhaps the authors could show some examples here of the different categories. Isn't there a third category with disappearing suprathermal electrons? The increase or decrease of the suprathermal electrons is given by the same parameter $\Delta n_{e,S}$ but it is not stated clearly here. (If there is a word and/or figure limit for the paper the authors can ignore my suggestions about including more plots)
- Line 145: ''…measured density outside the depletion.'' → Do the authors use a mean/median or just the first measurement outside the depletion? Could the authors state this in the text? This would be much easier to show if the authors included vertical lines in Figure 1 showing where the depletion starts and ends.
- Line 146: Same as the previous comment, now for $\Delta n_{e,S}$.
- Line 153: ''…discussed in previous studies of depletions…'' → Could the authors give some examples and include the corresponding references?
- Lines 151-154: The six events for which there is no enhancement in the suprathermal electrons, where are they located? Is their $\zeta$ larger? Are these events also included in Figure 5b and if so where exactly?
- Lines 158-159: ''…many events at low altitudes…crustal fields are stronger.'' → Perhaps this statement can be quantified somehow and the authors could provide some numbers to support it because I see also intermediate altitudes with high $\Delta n_{e,C}$ values.
- Lines 161-164: Perhaps the authors could elaborate on their results of Figure 6? Would it be useful if a ''depth'' is also defined for the ion depletions and be compared with the electron ones? Also why are there a few cases for which there is an enhancement in the ion densities? Where are these events observed?
- Line 188: ''…three events exhibit a reduction in ion temperatures…'' → It is difficult to see the three events in the Figure. A vertical dashed line at zero may help.
- Lines 189-190: ''Since we investigate…available temperature data decreases.'' → I am confused with that sentence. I am not sure what the authors want to say here.
- There are several statements in the paper about the number of events and how many events for different kinds of measurements are available. I was confused in the end. How many events were used for the analysis of the depletions, the suprathermal electrons and the ion temperatures?
- Figure 5: Would it be useful to color-code the altitude in Figure 5a as well?
- Figure 7: Could the authors add more lines below the plot, like the altitude and the SZA for example?

**5. Conclusion:**

- Lines 243-245: Could the authors elaborate in the main text (observations section) on how the parameters they use (the differences of ionospheric densities) minimize the effects mentioned here?

**TECHNICAL CORRECTIONS**

**Abstract:**

- Line 12: ''…the crustal magnetic field are…'' → ''…the crustal magnetic fields are…''

**1. Introduction:**

- Line 54: ''…depletions on the nigh side of Mars…'' → ''…depletions on the night side of Mars…''
- Line 54: ''…(Cao et al. 2022) argued…'' → ''…Cao et al. 2022 argued…''
- Line 57: ''…cooccurrence…'' → ''…co-occurrence…''

**2. Methodology, Event Selection, and Data Sources:**

- Line 77: ''…measurements between 2015 up to 2022 for ionospheric… → ''…measurements between 2015 and 2022 for ionospheric..'' or "…measurements from 2015 up to 2022 for ionospheric…''
- Lines 78-79: This sentence seems a little strange.

**3. Observations:**

- Line 125: ''In Figures 3 we discuss…'' → ''In Figure 3 we show…''
- Line 125-126: the verb is missing
- Line 144: ''…in cold or bulk electr on density'' → ''…in cold or bulk electron density''
- Line 162: ''…with increase in $\Delta n_{e,s}$…'' → ''…with increase in $\Delta n_{e,C}$…''

**4. Discussion:**

- Line 206: ''…which atomic oxygens becomes…'' → ''…which atomic oxygen becomes…''
- Line 209: ''…relevant reactions area listed below'' → ''…relevant reactions are listed below'' maybe also say reactions and reaction rates are listed below?
- Line 230: ''…removes caused…'' ?

**Figures:**

- In different parts of the paper the authors refer to the panels of the figures sometimes as Panel a, for example and sometimes as Panel (a). Perhaps they could use just one way.
- Figure 1: The lines below the plot of X, Y, Z are not aligned with the numbers.
- Figure 1: The y ticks labels on the first panel are too small compared with the other panels.
- Figure 5: The letters a) and b) above the panels are in different positions.
- Figure 6: ''Excluded in the figure are…''→''Eight depletion events… are excluded from the figure.''

---

## Author Comment (AC1)

RC1

General Comments

- Interesting approach, followed by an interesting and relevant discussion.

We would also like to thank the reviewer for taking the time to review this manuscript and providing useful comments which improved the manuscript. Please note that we have moved the figures to the manuscript text.

- I miss O+ also being cited in the Abstract.

We have modified the text to include O+ ion in line 13 of the abstract. If the reviewer is referring to $O^+$ mentioned in the second to last line of the abstract, that is because only $O^{2+}$ temperatures are currently available.

Scientific Questions

- Line 27: Explain "stand-off distance", or change it to another word. This may not be obvious to non-native English speakers, like myself, and can cause confusion even within the field of research.

We have removed the "stand-off" and changed the sentence to "*short distance of the bow shock boundary*"

- Lines 89-90: "In deriving Equation 1, effects of generic similarities between time series (constant arrays), singularities, and absolute strength of the fields are also considered." > How are these effects considered exactly?

We wanted to have a quantitative measure of the difference between magnetic fields observed by the spacecraft and the modeled fields at any given time. We began by simply comparing field strengths between the two sets, applied it to initially a small set of events. We noticed this simple difference approach gives small differences for small magnetic fields far away from the crustal fields, while larger differences for events on top of strong crustal fields but with slight deviations. That meant there should be a normalizing factor in the approach/model. We gradually modified the equation (trained the model) and expanded the event sets and monitored the model for robustness for all the events considered.

- Line 95: "The minimum accepted quality flag in LPW data is 50." > What does that mean, considering that the data never reached the density of 50 cm-3? What I interpreter from this is that the whole dataset should not be accepted? Or is it the opposite? Also, write the units after "50".

The LPW density and temperature data are reported with a quality/uncertainty flag which ranges from 0-100, regardless of the actual density and temperature values. This number among other factors depend on the quality of the fitting performed on the I-V curve to determine the density and temperature. This is the text included in each data file (the source may vary):

> *flag_info  Integers: The uncertainty of the values with a scale of 0-100. 100 is the best quality.  Use data with flag value above 50. #  The 5 digit decimal number represent a binary number containing information of specific atticue and sc activities. For all numbers see instrument SIS.*

> *flag_source What is used to evaluate the flag:  # MAVEN shadow information # MAVEN wake information # MAVEN thruster information.*

We have removed this sentence from the text as the information is include with the data and to avoid further confusion.

- Figure 1: Are you showing an example of electron depletion that is actually associated to regions of strong crustal magnetic fields? I can see from Panel a that the measurements of Bx and Bz are lower than the predicted (modeled) values. Probably the measured |B| will also be lower than the modeled |B|. How does that relate to your proximity parameter for this event? Is it considered high or low?

This event is a pass. The proximity parameter for this event is 1.16. The proximity parameter for all events considered is in the range of [0.19, 4.92], with a mean value of 2.18 and a median of 1.91. We have added this information to the text in line: 95

- Line 100: "down select": Do you mean you eliminate profiles with proximity parameter less than 5, or the opposite? "Down select" sounds ambiguous here, please use something like "eliminate", or rather "we select only", depending on what you mean.

We've replaced the word down select with, "We kept…".

- It would be interesting to see a map of crustal magnetic fields at Mars overlayed by the locations of the depletion events you analyse. I assume your proximity parameter could also catch events in the Northern hemisphere, far away from intense crustal fields.

That is correct. The crustal fields are strongest in the southern hemisphere. However, models show that there are weaker fields at other regions, and our manuscript is focused on events around any crustal field. The event shown at the bottom of this response letter shows the highest observed latitude event. And we are able to see matching patterns in the field components.

- Figure 2: Are Br, Btheta and Bphi values that are measured by the spacecraft or values from the crustal magnetic field model?

They are in-situ spacecraft measurements. We've added this clarification in line 135.

- Figures 3 and 4: Please insert a scale bar next to the histograms showing number/probability of events. Right now, the histograms are only qualitative.

We have added scale bars to all histograms in Figures 3 and 4.

- Line 129/Figure 3: "depletions are found more likely around crustal magnetic fields pointing eastward". What is the physical explanation for that? Does it have to do with the preferential direction of the solar wind magnetic field, perhaps?

The solar wind IMF, and the motional electric field could play a role. However, we don't have an upstream solar wind monitor to determine that. Additionaly, we did not restrict the events to a certain field orientation, or a specific region which could introduce a bias to the analysis. Except requiring proximity to crustal fields. We note that these structures are indeed intriguing and demand more attention from the community. For instance, one area that we will focus on in a follow up work is to identify areas of elongated stagnation.

- Line 131/Figure 4: "(i.e., exiting the Martian surface)". Why do you think we do not see as many events for $Br/|B| = -1$, i.e., when the field is entering the surface?

Similar to the comment above, we think it is worth noting what the analysis shows which is based on observations. We are however, unaware of a physical mechanism that gives preference to the field polarity.

- Line 161/Figure 6: "There is a shift towards higher O2+ variations with increase in $\Delta ne,s$ (the Colorbar)." What shift? Do you mean density variations of O2+ are larger than of O+? The sentence is confusing. Also, you say $\Delta ne,s$ but the colorbar shows $\Delta ne,c$.

The figure label is correct and we have modified the text to "$\Delta ne,C$". We have also reworded the confusing sentence to: "*There are higher variations in O2+ at lower altitudes with increasing $\Delta ne,C$ (shown with the color scale). That is because O2+ is the dominant ion species at low altitudes and has a shorter scale height compared to O+.*"

- Line 195: It lacks a sentence saying why we do not see even stronger variations in ion temperatures at altitudes higher than 400 km. Is it because the abundance of neutrals decreases? Or simply because there are not as many depletion events above 400 km?

Yes, indeed as you noted one reason is that there are not enough neutrals at higher altitudes. Another aspect to note here is that we focused on events around crustal fields. Magnetic fields act as conduit for electric potential. If there is frictional heating and an electric field driving ions at lower altitudes, such effects can "map" along the field lines and continue to impact ions at higher altitudes, although there could be other drivers in play as well. As the ion-neutral drift increases, the ion temperature also increases.

Technical Corrections

The manuscript contains several typos. Please, carefully proofread it and fix them.

Yes, thank you. We went through the entire text and made correction to the typos and a few sentence structures as necessary. These changes are marked up in the revised manuscript.

- Line 12: "crustal magnetic fields are..."

Fixed

- Line 24: "~200 km"

Fixed

- Line 27: "make"

Fixed

- Line 54: "night"

Fixed

- Line 54: (Cao et al., 2022) should be an inline citation.

Fixed

- Line 72: " in Section 2"

Fixed

- Line 79: "time series"

We think timeseries is a legitimate word and we made a change to use "timeseries" consistently throughout.

- Lines 78-79: "As such, we identified these events visually and when the density measurements exhibit sharp depletions both in timeseries and altitude profile data." > You identified the events both visually **and** when the density measurements exhibit sharp depletions? Is this correctly phrased?

We visually inspected both the timeseries and altitude profile for each event to make a selection. We have rephrased this sentence to clarify this point: "*As such, through visual inspection we identify events that exhibit sharp depletions in both the timeseries and altitude profile.*"

- Lines 91, 93, 96: You write Panel (a), Panel b, Panel c, and Panel d. Please, be consistent.

We have changed all instances to Panel a… etc.

- Line 94: "UTC"

Fixed

- Figure 1: The texts of X, Y, Z, and hhmm are not corresponding to the label rows.

We regenerated the figure with proper label spacing

- Line 125: "Figure 3"

Fixed

- Line 126: "... are shown below each plot."

Added "are shown". Thank you.

- Line 144: "electron"

Fixed

- Line 184: "ions and neutrals"

Added s to "neutral".

- Line 208: "due to"

Removed the redundancy.

- Line 209: "are listed"

Fixed

- Line 230: "removes caused"?

We have modified that sentence to: "*Electron-ion recombination removes both electrons and ions creating a density depression in the plasma.*"

---

## Author Response (AR1)

RC1

General Comments

- Interesting approach, followed by an interesting and relevant discussion.

We would also like to thank the reviewer for taking the time to review this manuscript and providing useful comments which improved the manuscript.

- I miss O+ also being cited in the Abstract.

We have modified the text to include $O^+$ ion in line 13 of the abstract. If the reviewer is referring to $O^+$ mentioned in the second to last line of the abstract, that is because only $O_2^+$ temperatures are currently available.

Scientific Questions

- Line 27: Explain "stand-off distance", or change it to another word. This may not be obvious to non-native English speakers, like myself, and can cause confusion even within the field of research.

We have removed the "stand-off" and changed the sentence to "*Lack of a global dipole field at Mars and the short distance of the bow shock boundary from the surface makes the ionosphere highly susceptible to upstream effects.*"

- Lines 89-90: "In deriving Equation 1, effects of generic similarities between time series (constant arrays), singularities, and absolute strength of the fields are also considered." > How are these effects considered exactly?

We wanted to have a quantitative measure of the difference between magnetic fields observed by the spacecraft and the modeled fields at any given time. We began by simply comparing field strengths between the two sets, applied it to initially a small set of events. We noticed this simple difference approach gives small differences for small magnetic fields far away from the crustal fields, while larger differences for events on top of strong crustal fields but with slight deviations. That meant there should be a normalizing factor in the approach/model. We gradually modified the equation (trained the model) and expanded the event sets and monitored the model for robustness for all the events considered.

- Line 95: "The minimum accepted quality flag in LPW data is 50." > What does that mean, considering that the data never reached the density of 50 cm-3? What I interpreter from this is that the whole dataset should not be accepted? Or is it the opposite? Also, write the units after "50".

The LPW density and temperature data are reported with a quality/uncertainty flag which ranges from 0-100, regardless of the actual density and temperature values. This number among other factors depend on the quality of the fitting performed on the I-V curve to

determine the density and temperature. This is the text included in each data file (the source may vary):

> *flag_info  Integers: The uncertainty of the values with a scale of 0-100. 100 is the best quality.  Use data with flag value above 50. #  The 5 digit decimal number represent a binary number containing information of specific atticue and sc activities. For all numbers see instrument SIS.*
>
> *flag_source What is used to evaluate the flag:  # MAVEN shadow information # MAVEN wake information # MAVEN thruster information.*

We have removed this sentence from the text as the information is include with the data and to avoid further confusion.

- Figure 1: Are you showing an example of electron depletion that is actually associated to regions of strong crustal magnetic fields? I can see from Panel a that the measurements of Bx and Bz are lower than the predicted (modeled) values. Probably the measured |B| will also be lower than the modeled |B|. How does that relate to your proximity parameter for this event? Is it considered high or low?

This event is a pass. The proximity parameter for this event is 1.16. The proximity parameter for all events considered is in the range of [0.19, 4.92], with a mean value of 2.18 and a median of 1.91. We have added this information to the text in line: 100

- Line 100: "down select": Do you mean you eliminate profiles with proximity parameter less than 5, or the opposite? "Down select" sounds ambiguous here, please use something like "eliminate", or rather "we select only", depending on what you mean.

We've replaced the word down select with, "We focus on density depletion events with $\zeta < 5$".

- It would be interesting to see a map of crustal magnetic fields at Mars overlayed by the locations of the depletion events you analyse. I assume your proximity parameter could also catch events in the Northern hemisphere, far away from intense crustal fields.

The event distribution map across the Martian surface has already been shown in Duru et al. (2023), Basuvaraj et al. (2022). In our study, we have focused on event near the crustal fields only. The reviewer is correct. The crustal fields are strongest in the southern hemisphere. However, models show that there are weaker fields at other regions, and our manuscript is focused on events around any crustal field. The event shown at the bottom of this response letter shows the highest latitude observed in an event.

- Figure 2: Are Br, Btheta and Bphi values that are measured by the spacecraft or values from the crustal magnetic field model?

They are modeled crustal magnetic field measurements. This comment was emphasized by both reviewers. We have performed this analysis using both modeled crustal fields and in-situ spacecraft measurements. But we decided to discuss the results in the context of modelled

crustal fields due to reasons described below. We've added the following clarification to the text: *" The magnetic field data in Figures 2, 3, and 4 are from the crustal field model. We also performed a similar analysis using spacecraft measurements of the magnetic field in the middle of each depletion. The events appear more evenly distributed along $\hat{B}_r$, $\hat{B}_\theta$ and $\hat{B}_\varphi$ when in-situ spacecraft data are used. In-situ magnetic field measurements include effects of upstream magnetic field as well as fields due to local currents which can change the modeled field orientation."*

- Figures 3 and 4: Please insert a scale bar next to the histograms showing number/probability of events. Right now, the histograms are only qualitative.

We have added scale bars to all histograms in Figures 3 and 4.

- Line 129/Figure 3: "depletions are found more likely around crustal magnetic fields pointing eastward". What is the physical explanation for that? Does it have to do with the preferential direction of the solar wind magnetic field, perhaps?

The solar wind IMF, and the motional electric field could play a role. However, we don't have an upstream solar wind monitor to determine that. Additionally, we did not restrict the events to a certain field orientation, or a specific region which could introduce a bias to the analysis. Except requiring proximity to crustal fields and ability to observe the ionosphere. We note that these structures are indeed intriguing and demand more attention from the community.

We have added the following statement to the text: *". The significance of such dependence and possible relationship to the planetary rotation will be investigated in future studies."*

- Line 131/Figure 4: "(i.e., exiting the Martian surface)". Why do you think we do not see as many events for $B_r/|B| = -1$, i.e., when the field is entering the surface?

Similar to the comment above, we think it is worth noting what the analysis shows which is based on observations. We are however, unaware of a physical mechanism that gives preference to the field polarity.

- Line 161/Figure 6: "There is a shift towards higher O2+ variations with increase in $\Delta ne,s$ (the Colorbar)." What shift? Do you mean density variations of O2+ are larger than of O+? The sentence is confusing. Also, you say $\Delta ne,s$ but the colorbar shows $\Delta ne,c$.

The figure label is correct and we have modified the text to "$\Delta ne,C$". We have also reworded the confusing sentence to: *"There are higher density variations in $O_2^+$ at lower altitudes with increasing $\Delta n_{e,C}$ (shown with the color scale). $O_2^+$ is the dominant ion species at low altitudes and has a shorter scale height compared to $O^+$."*

- Line 195: It lacks a sentence saying why we do not see even stronger variations in ion temperatures at altitudes higher than 400 km. Is it because the abundance of neutrals decreases? Or simply because there are not as many depletion events above 400 km?

Yes, indeed as you noted one reason is that there are not enough neutrals at higher altitudes. Another aspect to note here is that we focused on events around crustal fields. Magnetic fields act as conduit for electric potential. If there is frictional heating and an electric field driving ions at lower altitudes, such effects can "map" along the field lines and continue to impact ions at higher altitudes, although there could be other drivers in play as well. As the ion-neutral drift increases, the ion temperature also increases.

We have clarified this further by adding: *"Such a velocity difference typically increases with altitude, while the heating process itself is bound by the abundance of the local neutral species."*

Technical Corrections

The manuscript contains several typos. Please, carefully proofread it and fix them.

Yes, thank you. We went through the entire text and made correction to typos and a few sentence structures as necessary. These changes are marked up in the revised manuscript.

- Line 12: "crustal magnetic fields are..."

Fixed

- Line 24: "~200 km"

Fixed

- Line 27: "make"

Fixed

- Line 54: "night"

Fixed

- Line 54: (Cao et al., 2022) should be an inline citation.

Fixed

- Line 72: " in Section 2"

Fixed

- Line 79: "time series"

We have changed all the instances to "time series" consistently throughout.

- Lines 78-79: "As such, we identified these events visually and when the density measurements exhibit sharp depletions both in timeseries and altitude profile data." > You

identified the events both visually **and** when the density measurements exhibit sharp depletions? Is this correctly phrased?

We visually inspected both the time series and altitude profile for each event to make a selection. We have rephrased this sentence to clarify this point: "*Due to the highly dynamic and turbulent Martian ionosphere, our automated detection algorithm produced many false flags. As such, we identify these events through visual inspection. For each orbit leg, we inspect the electron density profile to identify an isolated depletion event showing the highest deviation from the general pattern of an exponentially decreasing profile (see the example in Figure 1).*"

Please also see our response to the second comment by the other reviewer below.

- Lines 91, 93, 96: You write Panel (a), Panel b, Panel c, and Panel d. Please, be consistent.

We have changed all instances to Panel a… etc.

- Line 94: "UTC"

Fixed

- Figure 1: The texts of X, Y, Z, and hhmm are not corresponding to the label rows.

We've regenerated the figure with proper label spacing

- Line 125: "Figure 3"

Fixed

- Line 126: "... are shown below each plot."

Added "are shown". Thank you.

- Line 144: "electron"

Fixed

- Line 184: "ions and neutrals"

Added s to "neutral".

- Line 208: "due to"

Removed the redundancy.

- Line 209: "are listed"

Fixed

- Line 230: "removes caused"?

*We have modified that sentence to: "Electron-ion recombination removes both electrons and ions creating a density depression in the plasma."*
* * *
**RC2**

**Ionospheric density depletions around crustal fields at Mars and their connection to ion frictional heating**

**GENERAL COMMENTS**

The authors investigate how density depletions in the ionosphere of Mars are correlated with the crustal fields and show how said depletions might be connected to ion frictional heating. The paper is well written and presents different aspects of the topic taking advantage of the available MAVEN observations from several instruments. The authors define certain parameters which then use to quantify the correlation between the crustal fields and the ionospheric depletions and their connection to ion frictional heating. The plots are clear and help the reader to understand the main points of the paper. There are some parts in the paper though that need further clarification.

*We would like to thank the reviewer for their useful comments which helped us improve our manuscript. Please note that line numbers below refer to the revised version of the manuscript with changes incorporated.*

**SPECIFIC COMMENTS**

**2. Methodology, Event Selection, and Data Sources:**

- Could the authors perhaps add a short paragraph about the instruments used in the paper? For example, what each instrument measures and such.

*The summary of what each instrument measures is already provided in Lines 100 – 115, and several other instances throughout the manuscript.*

- Do the authors survey the whole LPW data from 2015 to 2022 or a part of it? For example, only dayside, what range of SZAs and altitudes? or have the

authors used some other specific criteria? How many orbits do the authors check? Maybe the authors could be more specific here.

We surveyed the entire LPW data between 2015 and 2022. The number of orbits in this period is 15498 – 498 = 15000, with several criteria checked automatically. We have added the following description to the text to clarify the selection process better:
In Line 80-85: "*As such, we identify these events through visual inspection. For each orbit leg, we inspect the electron density profile to identify an isolated depletion event showing the highest deviation from the general pattern of an exponentially decreasing profile (See the example in Figure 1). We skipped orbits with no or limited data points where the structure of the ionosphere cannot be observed, orbits with periapsis at high altitudes (above 400 km), and orbits that showed high amplitude variations and multiple depletions and enhancements in the density profile.*"

And in Lines 267-270: "*Our survey of MAVEN-LPW ionospheric density profiles results in 1570 density depletion events. We then check the crustal field proximity condition on these events and obtain 242 events with the proximity parameter $\zeta < 5$. Events below this threshold seem to show a recognizable signature of crustal fields.*"

One issue that required extra attention was distinguishing the sudden density depletion events from ionopause crossings or variations near the ionopause. Depending on the orbit trajectory, and the ionopause boundary motions, some partial ionopause crossings could pass as sudden ionospheric density depletion.

- Line 79: Could the authors demonstrate an identification example plot from the data like they show in Figure 1. Perhaps the authors could just add to Fig. 1 a density profile and some lines on the plots indicating where the event starts and ends to show how they identify the depletions. That would also help the readers later when the authors describe the various Δn parameters and refer to measurements near the boundaries or out of the boundaries.

Per reviewer suggestion we have added a new panel to Figure 1 showing the altitude and SZA density profiles of electrons, O+ and O2+. We have also indicated the points where characteristic densities are selected.

- How do the authors search for the depletions? Do they first check the time series and if there is something they also check the profile? They check both time series and profiles for each orbit and compare? I would like to see a more detailed description of the identification process which would also fit nicely with the recommendation above about demonstrating the identification in a plot.

Please see our response to an earlier question above.

- When the authors calculate the proximity parameter $\zeta$, and use the magnetic filed measurements they should also state the coordinate system they use, for example that the $B_{i, sc}$ x, y, z components are in the MSO system.

Yes, the magnetic fields from the model and observations must be in the same coordinate

system (MSO coordinates). Otherwise, it would not make sense to calculate the difference between individual field components.

- Lines 89-90: ''In deriving Equation 1, effects of generic similarities between time series (constant arrays), singularities, and absolute strength of the fields are also considered'' ➜ Could the authors elaborate on that? Perhaps they could explain these effects in more detail and how they are considered.

In deriving Eq.1, we wanted to have a measure of closeness to any crustal field, regardless of its strength, so that is why the differences are normalized by the strength of the field.

- Line 95: ''The minimum accepted quality flag in LPW data is 50'' ➜ Could the authors elaborate on that? What does that mean exactly? It would be useful for readers who are not familiar with LPW data too.

The LPW density and temperature data are reported with a quality/uncertainty flag which ranges from 0 to 100. This is regardless of the actual density and temperature values. This number among other factors depend on the quality of the fitting performed on the I-V curves to determine the density and temperature. This is the text from the LPW data archives:

> *flag_info  Integers: The uncertainty of the values with a scale of 0-100. 100 is the best quality.  Use data with flag value above 50. #  The 5 digit decimal number represent a binary number containing information of specific atticue and sc activities. For all numbers see instrument SIS.*

> *flag_source What is used to evaluate the flag:  # MAVEN shadow information # MAVEN wake information # MAVEN thruster information.*

As we noted in our response to the other reviewer's comment, we have removed this sentence from the text as this information is already include with the archived data.

- Line 100: ''We down select...''➜ Could the authors elaborate on that? Events with $\zeta < 5$ are selected but out of how many and why? Why do they authors choose $\zeta < 5$? Arbitrarily? Can the authors provide a plot like a histogram/distribution with all the $\zeta$ measurements and the crustal fields values to show why they choose 5? Or a plot that shows $\zeta$ as a function of the strength of the crustal fields?

$\zeta$ increases for events farther away from the crustal fields. We examined several events with different $\zeta$ values and determined that for events with $\zeta < 5$, spacecraft measurements contain effects of crustal fields. For events with $\zeta > 5$, this influence decreases, may only appear in one component of the magnetic field but not the others, or spacecraft measurements may look nothing like the crustal fields.

- Lines 101-102: ''...visual inspection of several events...''➜ how many events are identified in total? How many were inspected?

*"Our survey of MAVEN-LPW ionospheric density profiles results in 1570 density depletion events."* We have added this information to the text in line 267. Also, please see our response to the second comment above.

- Lines 101-102:''…the total number of events… a reasonable sample size''➔ In my opinion this is not the right reason to select the right value for $\zeta$. If there were fewer events for example, would the authors select events with much higher $\zeta$ – and thus farther from crustal fields – just to have a sufficient number to do statistics?

As we mentioned in our earlier response, in our experience developing and working with the proximity parameter $\zeta$, events with $\zeta < 5$ seem to have a good exposure to the crustal field effects. Future studies can follow the methodology laid out in this work and expand the event set. Due to the lack of global dipole field and a real magnetosphere at Mars, defining a magnetopause boundary, like at Earth where there is a clear signature of boundary crossing, is not possible. Neither is determining the instance when the magnetic field from the magnetosheath/pile up region magnetic fields ends, and crustal fields begin. This is in fact a hybrid transition at Mars. We also note that the crustal fields at Mars though originate from subsurface, are not static, and can be moved around by upstream pressure pulses, or local currents. Therefore spacecraft measurements can deviate from the model predictions, and result in high $\zeta$ values. By using $\zeta$ we want to have a more quantitative measure of analyzing events near the crustal fields.

- It would be helpful for the readers if the authors could somehow present either by giving some numbers or with a plot as previously suggested, what happens at $\zeta = 5$. How strong the crustal fields are in the identified depletions for $\zeta = 5$, and what happens below and above this limit.

We believe that we have already addressed this comment in our earlier responses. Please see the response to the comments on "Line 100" and "Lines 101-102" above.

- Line 102:''We find 242 events in LPW data set.'' ➔ Total events? Events with zeta<5? Also, in what locations the authors see the depletions? Perhaps the authors can include a crustal field map with the locations of the depletions and/or altitude and SZAs of the events.

Thank you for this comment. We have rephrased these sentences to better demonstrate the event selection process and in accordance with our earlier responses as follows: *"This threshold is set by visual inspection of several events as we determined that for events with $\zeta < 5$, spacecraft measurements certainly contain contributions from crustal fields. The chosen threshold also limits the total number of events that would go into our statistical analysis to provide a reasonable sample size. 242 events met this criterion."*

- Figure 1➔ The authors could add the altitude in km and the SZA as well below the figure.

We have added a new panel to Figure 1 showing this information.

**3. Observations:**

- Lines 112-113: An example here would be helpful. If the authors could add vertical lines for example in Figure 1 with the minimum density and the limits of the hole (left and right) it would be easier to demonstrate exactly how they calculate the depth.

We have identified the event boundaries with blue dashed blue lines in Panel e of Figure 1.

- The authors now use spherical coordinates for the magnetic field. For the ζ calculation MSO was used. Maybe the authors could emphasize that and also elaborate a little on why spherical coordinates are more appropriate for their analysis.

For ζ calculations, the two fields can be in any identical coordinate system. In this section of the manuscript, we are discussing the crustal magnetic field orientation and a spherical coordinate aligned with planetary radius is more intuitive than the MSO coordinates.

- Line 120: "The depletion depth does not depend on ζ "➔ If I understand correctly, this means that the depletion depth does not depend on the crustal fields. Would be interesting to compare depletions farther from the crustal fields to see if you get the same depth distribution.

No, the ζ parameter is only a measure of proximity to the crustal field lines across the entire Martian surface. It is normalized by the strengths of the field strength, which removes the effect of crustal field strength. we want to study events that occur near crustal fields. We are not looking for finding a connection to the strength of fields. Further, and more important, with single point measurements, we can never know if the spacecraft measured the deepest point of the depletion or of the measurements are along the side of the depletion.

- Line 121: "...color coded by the total strength of the magnetic field." ➔ Do the authors mean the total strength measured or modelled (crustal fields) ?

They are colored by the total strength of the modeled crustal fields. We've added the word strength to the end of this sentence: "*Data points on this panel are color coded by the magnetic field strength.*"

We performed this analysis using in-situ measurements of the magnetic field as well. The correlations and patterns appeared smoother. We have added the following description to the text in Lines 148-152, to clarify these points:

"*The magnetic field data in Figures 2, 3, and 4 are from the crustal field model. We also performed a similar analysis using spacecraft measurements of the magnetic field taken at the middle of each depletion. The events appear more evenly distributed along $\hat{B}_r$, $\hat{B}_\theta$ and $\hat{B}_\varphi$ when in-situ spacecraft data are used. In-situ magnetic field measurements include effects of upstream magnetic field as well as fields due to local currents which can change*

*the modeled field orientation.*"

■ Line 121-123: "Events with the highest magnetic...appear at low ζ...mostly above 0.1"➜ I am a little confused with that statement. Events with high magnetic field strength would also be the events where crustal fields dominate so by definition ζ should be low.

That is correct. We simply pointed out the observations of the cluster of data points at very high magnetic field strength. As the reviewer noted, this behavior is expected. To respond to this comment and another related comment further below, we have modified this sentence as follows: "*Several events with relatively high magnetic field strength (purple data points) appear at ζ < 2 (i.e., fields are mostly dominated by strong crustal fields). Though, as Equation 1 indicates, ζ is normalized by the strength of the field components and is only a measure of the proximity of the event and spacecraft to any crustal field regardless of the field strength.*"

■ Also the fact that the depletion depth is larger for those events does it mean that there is after all a correlation between crustal fields and depletion depth? Because the previous statement says that the depth does not depend on ζ.

Please note that we define the depletion depth as the density ratio of lowest density point within the depletion to the ambient density. So smaller ratios mean deeper holes or "larger depth".

■ Figure 3 – Perhaps the authors could give a more detailed description of the plot. Since the same format is used in Figure 4 as well.

We have modified the caption of both figures 3 and 4. Details about Figure 3 is also discussed in Lines 135 – 140.

■ Line 133: No correlation between ζ and altitude. So no correlation of the crustal fields and the altitude? Perhaps the authors could emphasize that in the paper.

Please see our response to the related comment above "Line 121-123: ". Also note Figure 4 shows the normalized component (Br / |B|) and not Br. It appears the reviewer is referring to the relationship between magnetic field strength and altitude. However, that is not what we discuss in this section. Since we normalize the differences by the field strength, any correlation between field strength and consequently altitude will be removed. We have deleted this sentence from text as the point is mute.

■ Lines 141-143: "some depletion events...or near the boundaries" ➜ Perhaps the authors could show some examples here of the different categories. Isn't there a third category with disappearing suprathermal electrons? The increase or decrease of the suprathermal electrons is given by the same parameter $\Delta n_{e,S}$ but it is not stated clearly here. (If there is a word and/or figure limit for the

paper the authors can ignore my suggestions about including more plots)

We have rephrased these sentences to clarify that we are referring to as enhancements: *"Increase of suprathermal electrons within depletion events can be through an increase in the flux of suprathermal electrons over almost the entire event period, or via a short beam-like surge of electrons either in the middle of the depletion or near the boundaries."*

■ Line 145: "…measured density outside the depletion." ➔ Do the authors use a mean/median or just the first measurement outside the depletion? Could the authors state this in the text? This would be much easier to show if the authors included vertical lines in Figure 1 showing where the depletion starts and ends.

We use the average of the measurements on two sides/edges of the depletion as a measure of the outside density. We have added the following line to the text to address this issue: *"The outside (ambient plasma) density measures are the maximum of the density measurements at both edges of the depletion (see Panel e in Figure 1)."*

■ Line 146: Same as the previous comment, now for $\Delta n_{e,S}$.

Please see the comment above.

■ Line 153: "…discussed in previous studies of depletions…" ➔ Could the authors give some examples and include the corresponding references?

References which show example cases were already included in line 153-154. We've added Duru et al. 2023 to these references.

■ Lines 151-154: The six events for which there is no enhancement in the suprathermal electrons, where are they located? Is their ζ larger? Are these events also included in Figure 5b and if so where exactly?

The Y-axis of the two panels shows two different parameters. On Panel a, it is the difference between the maximum suprathermal electron density within the depletion and the density of suprathermal electrons in ambient plasma. Panel b shows the difference between maximum suprathermal electron density and average suprathermal electron density within the depletion and is always a positive quantity.

■ Lines 158-159: "…many events at low altitudes…crustal fields are stronger." ➔ Perhaps this statement can be quantified somehow and the authors could provide some numbers to support it because I see also intermediate altitudes with high $\Delta n_{e,C}$ values.

We have modified the text in these lines as follows: *"Events at low altitudes (cyan colors), where ionospheric densities are typically higher and crustal fields stronger, exhibit high values of $\Delta n_{e,C}$ (> $10^3$ $cm^{-3}$) compared to events at other altitudes. $\delta n_{eS}$ for these events also increases by at least a factor of 2, suggesting that beams of suprathermal electrons are likely to occur at lower altitudes where crustal fields are stronger."*

Please see our response to the comment on "Line 133:" above.

- Lines 161-164: Perhaps the authors could elaborate on their results of Figure 6? Would it be useful if a ''depth'' is also defined for the ion depletions and be compared with the electron ones? Also why are there a few cases for which there is an enhancement in the ion densities? Where are these events observed?

The comparison of depletion depth for ions versus electrons has already been done by Basuvaraj et al. (2022), Figure 5b. We have added the following sentences emphasizing the physical mechanism we discuss later in the text: *"The strong correlation observed between the depletion of cold electrons and ions in the ionosphere indicates that a similar physical mechanism is in play removing electrons and all ions from the ionosphere. The likely candidate mechanism would be ion-electron recombination which consumes both electrons and ions."*

- Line 188: ''…three events exhibit a reduction in ion temperatures…''➔ It is difficult to see the three events in the Figure. A vertical dashed line at zero may help.

We used the actual arrays to determine the number of negative elements and there were three of them.

- Lines 189-190: ''Since we investigate…available temperature data decreases.'' ➔ I am confused with that sentence. I am not sure what the authors want to say here.

Thanks for this comment. We have modified the manuscript to convey these two points: *"As shown in Figure 8, ion temperature inside most depletion events show an increase. In some event the ion temperatures at the center of the depletion where the ion densities are at a minimum, are not well determined. Depletion and reduced counts of ions within these events undermines the reliable determination of ion temperatures. Figure 8 also indicates …"*

- There are several statements in the paper about the number of events and how many events for different kinds of measurements are available. I was confused in the end. How many events were used for the analysis of the depletions, the suprathermal electrons and the ion temperatures?

The number of events available in different instrument datasets were mentioned in lines 106-112. So we did not change the text in the paragraph. We have added a few more sentence to the conclusion section defining the total number of events. Please also see our response to the second comment above.

- Figure 5: Would it be useful to color-code the altitude in Figure 5a as well?

It does not reveal any new information. It shows similar information as in Panel b, and no dependence on $\Delta n_{e,S}$. Please note that we have modified the statements in these lines in accordance to our response to the comment on "Lines 158-159".

■ Figure 7: Could the authors add more lines below the plot, like the altitude and the SZA for example?

Yes, we have added SZA and altitude labels to this figure.

**5. Conclusion:**

- Lines 243-245: Could the authors elaborate in the main text (observations section) on how the parameters they use (the differences of ionospheric densities) minimize the effects mentioned here?

As has been done in previous work cited in the paragraph, instead of absolute density values, we look at the local changes in plasma conditions from the ambient conditions due to the structures. We have added the following description to the manuscript: *"To avoid and minimize density variations due to the solar zenith angle, seasons, and heliocentric distance on our analysis, instead of absolute density values, we compare the differences of the ionospheric densities between inside and outside the depletion events to quantify these structures (Andrews et al., 2023). We however note that a limitation of single point measurements is that depending on the spacecraft trajectory and path through a three-dimensional density structure, the observed density variations may not reflect the actual structure depth."*

**TECHNICAL CORRECTIONS**

**Abstract:**

- Line 12: ''…the crustal magnetic field are…''➜ "…the crustal magnetic fields are…'' Fixed.

**1. Introduction:**

- Line 54: ''…depletions on the nigh side of Mars…''➜''…depletions on the night side of Mars…'' Fixed.
- Line 54: ''…(Cao et al. 2022) argued…''➜''…Cao et al. 2022 argued…'' Fixed.
- Line 57:''…cooccurrence…''➜''…co-occurrence…'' Fixed.

**2. Methodology, Event Selection, and Data Sources:**

- Line 77: ''…measurements between 2015 up to 2022 for ionospheric… ➜ ''…measurements between 2015 and 2022 for ionospheric..'' or "…measurements from 2015 up to 2022 for ionospheric…'' Fixed.
- Lines 78-79: This sentence seems a little strange.

Fixed. We have rephrased the sentence to: *Due to the highly dynamic and turbulent Martian ionosphere, our automated detection algorithm produced many false flags.*

**3. Observations:**

- Line 125: "In Figures 3 we discuss…''➜ "In Figure 3 we show…" Fixed.
- Line 125-126: the verb is missing Fixed.
- Line 144:''…in cold or bulk electr on density''➜ ''…in cold or bulk electron density''

Fixed.

- Line 162: ''…with increase in $\Delta n_{e,S}$…''➔''…with increase in $\Delta n_{e,C}$…'' Fixed.

**4. Discussion:**

- Line 206: ''…which atomic oxygens becomes…''➔''…which atomic oxygen becomes…'' Fixed.
- Line 209: ''…relevant reactions area listed below''➔ ''…relevant reactions are listed below'' maybe also say reactions and reaction rates are listed below? Fixed and added "rates".
- Line 230: ''…removes caused…'' ? We have changed the sentence to: "*Electron-ion recombination removes both electrons and ions creating a density depression in the plasma.*" Thanks.

**Figures:**

- In different parts of the paper the authors refer to the panels of the figures sometimes as Panel a, for example and sometimes as Panel (a). Perhaps they could use just one way. We have changed all instances to "Panel a".
- Figure 1: The lines below the plot of X, Y, Z are not aligned with the numbers. This has been fixed.
- Figure 1: The y ticks labels on the first panel are too small compared with the other panels. We have increased the font for that panel.
- Figure 5: The letters a) and b) above the panels are in different positions. We've moved the label to a right position.
- Figure 6: ''Excluded in the figure are…''➔''Eight depletion events… are excluded from the figure.'' Fixed.

[Figure]